# Biopsychosocial Intervention for Stroke Carers (BISC): protocol for a feasibility randomised controlled trial (RCT)

Marion F Walker,[1] Shirley A Thomas,[1] Phillip J Whitehead,[1] Laura Condon,[1] Rebecca J Fisher,[1] Eirini Kontou,[1] Penny Benford,[1] Christine Cobley[2]

[1]Division of Rehabilitation and Ageing, University of Nottingham, The Medical School, Queen's Medical Centre, Nottingham, UK
[2]Department of Clinical Psychology, Derbyshire Healthcare NHS Foundation Trust, Derby, UK

**Correspondence to**
Professor Marion F Walker;
marion.walker@nottingham.ac.uk

## ABSTRACT

**Introduction** Reducing length of hospital stay for stroke survivors often creates a shift in the responsibility of care towards informal carers. Adjustment to the caregiving process is experienced by many carers as overwhelming, complex and demanding and can have a detrimental impact on mental and physical health and well-being. National policy guidelines recommend that carers' needs are considered and addressed; despite this, few interventions have been developed and empirically evaluated. We developed a biopsychosocial intervention in collaboration with carers of stroke survivors. Our aim is to determine whether the intervention can be delivered in a group setting and evaluated using a randomised controlled trial (RCT).

**Methods and analysis** Feasibility RCT and nested qualitative interview study. We aim to recruit up to 40 stroke carers within 1 year of the stroke onset. Carers are randomised to usual care or usual care plus biopsychosocial intervention. Each intervention group will consist of five stroke carers. The intervention will focus on: psychoeducation, psychological adjustment to stroke, strategies for reducing unwanted negative thoughts and emotions and problem-solving strategies. The main outcome is the feasibility of conducting an RCT. Carer outcomes at 6 months include: anxiety and depression, quality of life and carer strain. Data are also collected from stroke survivors at baseline and 6 months including: level of disability, anxiety and depression, and quality of life.

**Ethics and dissemination** Favourable ethical opinion was provided by East Midlands – Nottingham2 Research Ethics Committee (14/EMI/1264). This study will determine whether delivery of the biopsychosocial intervention is feasible and acceptable to stroke carers within a group format. It will also determine whether it is feasible to evaluate the effects of the biopsychosocial intervention in an RCT. We will disseminate our findings through peer-reviewed publications and presentations at national and international conferences.

**Trial registration number** ISRCTN15643456; Pre-results.

## INTRODUCTION
### Background and rationale

A carer has been defined as 'a person of any age who provides unpaid help and support to a relative, friend or neighbour who cannot manage to live independently without the

---

### Strengths and limitations of this study

► Enhancing the well-being of carers is a national priority. However, few interventions for carers have been developed or evaluated.
► This is a pragmatic trial conducted in a real-world setting. The intervention content is based on the findings from the literature, developmental work with carers of stroke survivors and stroke rehabilitation experts.
► This feasibility study will be conducted in a single site only.
► The intervention focuses only on the initial stages of carer support (up to 1 year poststroke onset). Significant problems may develop for carers at later stages that need to be identified and referred for more intensive/specialist support.

---

carer's help due to frailty, illness, disability or addiction'.[1] Carers play a vital role in the early rehabilitation process and long-term management of the stroke survivor.[2] Carers deal with a range of care needs and demands including mobility, self-care, communication difficulties as well as cognitive impairment, mood and personality changes in the stroke survivor.[3]

The latest figures from the Sentinel Stroke National Audit in England, Wales and Northern Ireland show that the median length of inpatient stay is between 7 and 8 days; however, just under one-third of stroke survivors who were discharged requiring help with daily activities received assistance from informal carers.[4] This demonstrates that increasingly shorter hospital stays coincide with an earlier transfer of care to informal carers in the community. The Care Act 2014 has placed a responsibility on local authorities in England to consider the well-being of carers as being of equal importance to the well-being of the people they care for.[5] The importance of providing support and intervention to carers has also been emphasised in national stroke guidelines.[6 7] Consequently,

it is becoming more urgent to develop appropriate and effective interventions to meet the specific needs of carers of stroke survivors.

A growing number of people are unexpectedly finding themselves in the caregiving role. Although it can be a positive and rewarding role,[8] the increased demands associated with informal caring can place carers at elevated risk of poorer mental and physical health, accompanied by reduced opportunity for paid employment and social interaction and activity.[3] An estimate of the psychosocial impact that might be associated with stroke care, drawn from a survey of carers,[9] shows that carers may experience: anxiety (79%), frustration (84%), sleeping disturbances (60%), depression (56%) and stress (57%). Deterioration in the health and well-being of the carer has important implications on the outcomes of stroke survivors including: poorer rehabilitation outcomes; reduced quality of life; heightened levels of depression; greater risk of mortality; poorer treatment adherence; and increased likelihood of being placed into institutional care, which has important cost implications for the NHS, social care services, the stroke survivor and their family.

An increased focus on the needs of stroke carers has led to a spate of recent systematic reviews including quantitative and interventional studies,[10–13] qualitative research[8 14] and economic evidence.[15] However, the results are equivocal and limited. Thus, there is no clear and robust evidence regarding the most effective and cost-effective interventions for stroke carers. This lack of evidence is not due to the lack of research in this field[11]; interventions directed at both stroke survivors and carers form the largest body of research and have predominantly focused on examining new models of service delivery,[16] such as care-giver training, the stroke family support worker[17 18] and multidisciplinary hospital and community stroke teams. These interventions however have predominantly focused on the stroke survivors' needs rather than the carers' needs,[19] and thus the needs of carers have largely been neglected. The few interventions to date that have been developed specifically for carers include education and information,[20] skills training[21] and social support.[22] Such interventions however have produced inconclusive findings, arguably because such interventions are failing to address and meet the specific needs of carers. Forster *et al*[23] evaluated a structured caregiver training programme delivered in hospital by multidisciplinary teams from stroke units. There was no difference between the intervention and usual care, and they concluded that the immediate period after stroke might not be the best time to deliver such a programme.

Given the high prevalence of psychological morbidity within the stroke carer population, there is likely to be a high demand for psychologically informed interventions targeted at informal carers of stroke survivors beyond the initial period of hospitalisation. Although evidence-based treatments for psychological difficulties exist, the associated costs and expenses of service delivery are high, with demand for treatment exceeding service capacity,

resulting in long waiting lists[24] and limited access.[25] There are other barriers experienced by carers wishing to pursue and access mental health services.[26] These barriers include a lack of attention by health professionals of the difficulties associated with the caregiving role and that general practitioners are often more likely to offer practical rather than psychological support. Together these reasons make it increasingly difficult for informal carers to access evidence-based psychologically informed interventions.

The biopsychosocial model of health and illness, as proposed by Engel,[27] suggests that psychobiological vulnerability is influenced by an interaction of biological (physical health), psychological (thoughts, emotions and behaviours) and social (relationships and roles) factors. The model emphasises the need for interventions to focus on both symptom reduction and on relapse prevention.[28] Psychological models such as cognitive–behavioural and interpersonal therapy have been deemed too fragmented and reductionist,[29] given that they do not integrate the biological and psychological factors, as well as social, environmental and stress factors that are known to interfere with psychological functioning.

There have been movements towards the use of biopsychosocial interventions for the treatment of psychological difficulties among the general population. However, evidence suggests that significant adaptations to such interventions are required prior to application to different clinical populations. Indeed, mental health services for carers have been criticised for not being tailored to address the unique and specific difficulties experienced by stroke carers.[30 31] Such difficulties can include having to manage the physical and cognitive impairment and behavioural difficulties the stroke survivor may be presenting with.[32 33] There is growing recognition of the importance of understanding carer's experiences when dealing with health resources and healthcare policy.[34] A systematic review of psychosocial interventions for stroke carers concluded that more randomised controlled trials (RCTs) of psycho-education programmes are needed.[12]

Considering this recommendation in the context of the wider literature on stroke carers, we developed a new biopsychosocial intervention specifically targeted at informal carers of stroke survivors. The intervention was developed collaboratively with stroke carers and designed to be delivered in a group format to offer participants the opportunity to meet and interact with people and listen to how others have coped. Delivering the intervention in a group format is also likely to be more time and cost efficient, which would be important given the current demand for psychological therapies.

This study is examining the feasibility of conducting a RCT to examine the effectiveness and acceptability of this group biopsychosocial intervention for stroke carers in the first-year poststroke.

## Research aim and objectives

The ultimate aim of this study is to evaluate whether a biopsychosocial intervention can improve psychological

outcomes in carers of stroke survivors (in the 1-year post-stroke period). However, we are not able to complete a definitive, powered trial until we have collected further information to inform the design of such a study. The purpose of this feasibility trial is to explore whether the biopsychosocial intervention for carers of stroke survivors is feasible and acceptable and to estimate the parameters for conducting a fully powered trial.

## Primary objective

The primary objective of this feasibility trial is to evaluate whether it is feasible to deliver a biopsychosocial intervention to carers of stroke survivors as part of an RCT.

## Secondary objectives

This feasibility RCT will test the integrity of the study protocol, such as the methods of data collection, randomisation procedures and the masking of independent assessors. This feasibility study will answer the necessary questions to inform a definitive multicentre trial that include:

► Can we identify participants willing to be randomised?
► Can we deliver the intervention as planned?
► Is the intervention acceptable to participants?
► Can we retain participants in the study?
► What are the most relevant outcome measures?
► What is the consent rate?

## METHODS AND ANALYSIS
### Study design and setting

This is a single-centre feasibility RCT with nested qualitative interview study. The RCT is a parallel group, two-arm trial with a 1:1 allocation ratio of biopsychosocial intervention:usual care control.

## Participants

Participants are carers of people who have had a stroke (stroke survivors). Our definition of a carer is a family member or friend who is/will be providing support for a stroke survivor who would not be able to manage without their help due to their condition. Carers will be recruited along with stroke survivors from stroke units at a university hospital, community stroke services and third sector stroke clubs and support groups. However, only the carer will receive the intervention; we will recruit the stroke survivor because we also aim to collect baseline and follow-up data from them.

The inclusion criteria are as follows:
Stroke carers
► aged 18 years or over
► carer of a person with a confirmed diagnosis of stroke within 1 year of stroke onset
► capacity to provide informed consent
► willing to attend a 6-week group intervention programme.
Stroke survivors
► aged 18 years or over
► confirmed diagnosis of stroke
► within 1 year of stroke onset

► capacity to provide informed consent or consultee opinion that the person would wish to participate.
The exclusion criteria are as follows:
Stroke carers
► unable to speak English
► engaged in other research involving biopsychosocial/psychological interventions
► people with visual (blindness) or auditory (deafness) impairments that would preclude them from participating in the therapy sessions.
Stroke survivors
► unable to speak English
► people engaged in other research involving biopsychosocial/psychological interventions.

## Intervention development

The intervention was developed based on the biopsychosocial model of health and illness[27] with the aim to address biological, psychological and social factors and symptom reduction and relapse prevention. The content was informed through a review of the literature and a series of focus groups conducted with carers. Thematic analysis of focus group data revealed specific difficulties and challenges experienced by carers in the early poststroke aftermath, and helpful coping strategies commonly used. We also conducted a nominal group approach with stroke rehabilitation experts to further refine the intervention. The biopsychosocial intervention was designed to recognise and target the difficulties commonly experienced by informal carers (identified through the focus groups and from the stroke literature). Additionally, helpful coping strategies used by the informal carers were used to further inform and adapt the content of the intervention. More detailed information about the development of the intervention will be provided in a further publication.

## Intervention and comparator

Participants are randomised as dyads. Participants will be randomised to either:
► Control group: usual care. Carers randomised to the control group will receive the usual range of routine care and services available to them. They will not receive the biopsychosocial intervention.
► Intervention group: biopsychosocial intervention, plus usual care
Carers randomised to the intervention group will receive the biopsychosocial intervention, in addition to usual care. The stroke carers randomised to receive the intervention will receive a 2-hour session once a week for 6 weeks. The time point at which the intervention will start will be agreed with the carer, in conjunction with other carers likely to be part of that group. This will occur when the stroke survivor they care for has been discharged from hospital, up to 1 year poststroke. We will aim to deliver the intervention to groups of approximately five people. However, in the event that it is not possible to coordinate sufficient people, or where carers are unable to attend the

group sessions, we will deliver the intervention on a one-to-one basis or in smaller groups. We will record this as part of our feasibility. The intervention will be delivered at a suitable venue, with sufficient space and access for carer group members. The intervention sessions will be facilitated by a research psychologist who has received training in the principles of biopsychosocial theory as well as specific training from members of the research team responsible for the development of the intervention. Clinical supervision and debriefing sessions will be provided by an experienced community mental health nurse with significant community stroke team experience and/or a clinical psychologist. Each session will last approximately 2 hours and will include a 15 min tea/coffee break that will allow participants to interact more informally with one another, and the session will conclude with a 15 min relaxation exercise. We anticipate that in the definitive trial, the intervention could be delivered by assistant psychologists with supervision from clinical psychologists.

The intervention programme is focused on adjustment to stroke, provision of psychoeducation and psychological support. The group programme is based on the principles of the biopsychosocial model. The sessions are designed to teach individuals to identify and use skills to reduce current and future distress, thus aiding coping and adjustment to the impact of stroke and their role as a carer. The sessions are also intended to increase awareness of the role of thoughts, emotions and behaviours and their influence on each other. By practising problem-solving and stress-management strategies, it is hoped that carers will experience fewer difficulties with their mood in the future.

For each session, there will be a presentation containing information about a topic and exercises to aid discussion. Sessions will be presented on Microsoft PowerPoint, and all participants will receive either electronic or paper copies of the slides as appropriate, accompanied by the exercise and an in-between session task. The topics will cover, for example, an introduction to stroke and caring, adjustment and mood, how to handle negative emotions and thoughts, dealing with problems and a well-being relapse prevention plan (which provides a set of coping mechanisms to deal with individual triggers of the stress response in relation to the role as stroke carers to encourage and foster positive mental health). The content of these sessions was informed by the findings of earlier work described above. Relaxation exercises at the end of each session will allow participants to feel calm and relaxed before finishing their session and will also be an effective tool for them to use outside of the session when experiencing high levels of anxiety or distress. Between session tasks will be set to encourage participants to practise exercises from the sessions in their own time. Participants who are identified to be experiencing significant issues that are out of the scope of

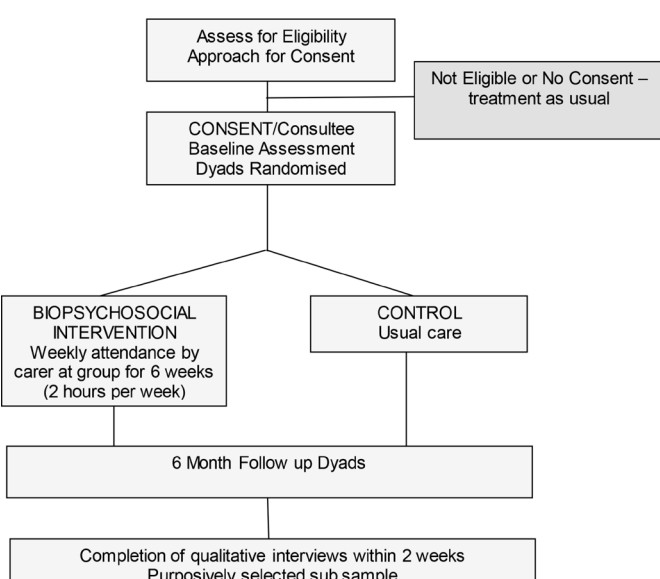

**Figure 1** Flow of participants through the study.

the intervention will be referred onto the appropriate specialist service, subject to their consent.

### Outcomes

The main outcome for the study is to determine the feasibility of conducting a larger, powered study. This will be a composite of: whether the eligibility criteria are realistic; whether stroke survivors and carers are willing to be randomised; the study attrition rate; the feasibility and acceptability of delivering the intervention; the suitability and sensitivity of outcome measures; and the most suitable outcome measure for use in the main study.

The stroke carer outcomes to be assessed at 6 months postrandomisation will be: anxiety and depression, health-related quality of life and carer strain. The outcome measures that will be used are: Hospital Anxiety and Depression Scale,[35] EuroQol 5D-5L[36] and the Caregiver Burden Scale.[37]

The stroke survivor outcomes to be assessed 6 months postrandomisation will be: level of disability, ability to perform personal activities of daily living, level of anxiety and depression and health-related quality of life. The outcome measures that will be used are: Modified Rankin Scale,[38] Barthel Index,[39] Hospital Anxiety and Depression Scale[35] and EuroQol 5D-5L.[36] The timeline and proposed flow of participants through the study is shown in figure 1.

### Feedback interviews

Qualitative semistructured interviews will be conducted with carers in both arms of the trial, within 2 weeks of their final outcome assessment. Up to 10 interviews will be completed with stroke carers in each arm. Our aim is to obtain feedback on all aspects of the study in addition to the intervention procedures, assessments, intervention (if received) and perceived outcomes. For those in the control group, the interviews will provide confirmation of the nature of usual care received. Participants will be purposefully selected to include carers of stroke

survivors with varying severity of stroke, age and gender. The interviews will be conducted by a researcher who had no involvement in the intervention delivery, thereby reducing social desirability response bias. The researcher conducting the interviews will become aware of the group allocation during the interview and so will not be masked to the intervention. These interviews will be audio recorded using a digital recorder, transcribed and analysed using a thematic analysis (following the procedure described by Braun and Clarke[40]). The interviews with participants will provide information feedback on their perception of progress over time and for those in the intervention group, the quality of the intervention provided, and as such will serve as a process measure. Insights from the qualitative data and analysis will serve to inform developments of the intervention programme in the future and to generate user-orientated proposals about areas for further investigations. This information will also inform us of any refinements to be made to the study procedures. An interview will also be conducted with the group facilitator after they have completed all therapy. This interview will ask about the ease of delivery of the intervention according to the manual and any challenges.

### Sample size, recruitment strategy, randomisation and blinding

For a feasibility study, no formal sample size calculation is required. The aim is to recruit up to 40 dyads (20 in each arm of the trial) to test the randomisation process and the feasibility of the study processes of delivering the intervention. This target should allow us to collect sufficient information on the suitability and sensitivity of the outcome measures for use with this population and the SD of the measures to inform a sample size calculation for a definitive trial. The median sample size for UK feasibility trials has been reported at 36,[41] which is broadly consistent with the planned target.

The trial opened for recruitment on 1 November 2015 and will close on 31 July 2017 or when 40 dyads have been recruited (whichever is soonest). Participants will be enrolled into the study by a member of staff from the Clinical Research Network or a member of the research team. The process for obtaining participant informed consent will be in accordance with the Research Ethics Committee guidance, Good Clinical Practice and any other regulatory requirements that might be introduced. Following a full explanation of the study, the participant will be required to provide informed written consent before they can participate. Where a consultee is required for a stroke survivor, the consultee shall provide a recommendation as to whether they consider the person would have agreed to take part in the study had they still had capacity to state their own preference. They will sign the consultee declaration should they believe that person would have wished to take part in the study.

Participants will be randomised at baseline following consent and completion of the baseline assessments. Randomisation to each group will be on a 1:1 basis—intervention:control. A simple randomisation procedure will be provided and overseen by the East Midlands Research Design Service. The group facilitator will be informed of group allocation as they will be providing the treatment. We will take every step to minimise allocation and outcome bias.

Trial participants will not be masked to group allocation because they will need to be informed as to whether they have been allocated to the intervention group receiving the biopsychosocial intervention or the control group. The participants' names, trial identifier numbers and treatment allocation will be stored on a password-protected database held by the group facilitator. This database will be used to allow treatment allocations to be identified at the end of the study.

Baseline data will be collected, and baseline assessments will be completed prior to randomisation. Baseline information will include:

1. demographic details including age, gender, ethnicity and employment
2. levels of anxiety and depression (Hospital Anxiety and Depression Scale)
3. quality of Life (EuroQol)
4. carer strain (Carer Burden Scale).

In addition, we will collect the following information from the stroke survivor and/or their medical notes (with consent):

1. stroke characteristics
2. language and cognitive abilities (Montreal Cognitive Assessment)
3. personal activities of daily living (Barthel Index)
4. stroke severity (National Institute of Health Stroke Scale)
5. quality of life (EuroQol)
6. which service (if any) the stroke survivor is discharged to (eg, Early Supported Discharge and intermediate care).

Follow-up assessment visits will be completed at 6 months postrandomisation by a research assistant who is masked to allocation. To minimise the risk of unmasking, prior to each contact, the participant will be reminded that the researcher who is to conduct their follow-up assessment is masked. It is possible that participants may reveal their group allocation to the outcome assessors and any instances of this will be recorded by researchers as part of the assessment of feasibility; researchers will also be asked to make their 'best guess' as to the group allocation of the participants to determine whether masking was successful. Other members of the research team and investigators will not be masked to group allocation for the purpose of managing the trial and delivering the interventions. It will not be possible to mask participants.

### Data collection, management and analysis

Data will be collected on a paper case report form (CRF) and will subsequently be entered onto a secure, password-protected, purposely designed electronic database. Each participant will be assigned a trial identity code

number, allocated at randomisation, for use on CRFs, other trial documents and the electronic database to ensure confidentiality. The documents and database will also use their initials and date of birth. CRFs will be treated as confidential documents and held securely in accordance with regulations.

When data collection is complete, a data quality check will be conducted in duplicate by two researchers, and a 10% sample of the database will be checked against the original paper CRF. Steps will be taken to minimise missing data by personal contact throughout the study period from the investigator, and every attempt will be made to locate participants for follow-up. Where participants are unavailable for follow-up, details of the attempts to contact them will be recorded. Outcome data will be collected in person, in the participant's home, by a research assistant to minimise the amount of missing data. For each outcome measure used where data are missing, an imputed average will be used for items where less than 10% of the overall measure is missing. Where more than 10% of a measure is missing, the entire measure will be coded as missing, unless the scoring criterion for that measure stipulates an alternative approach. We will not collect any further data for participants who withdraw from the study, but we will retain all data collected up until the point of withdrawal.

The following procedures will apply to data analysis.

### Acceptability of the study design

Descriptive statistics will be presented for the following feasibility outcomes: recruitment rates, proportion of carers screened who are eligible for enrolment and who provided consent, how easily carers can be identified, who met the criteria for the study, number of people who accepted intervention to take part in the RCT and number of individuals who attended the intervention/number of sessions they attended. The feedback interviews will provide further information regarding the acceptability of the intervention. Qualitative thematic analysis will provide an insight into carer perspectives of their experience of caring and what effect they think the intervention itself may have had (for the treatment group).

### Feasibility of completing the intervention

Proportion of carers completing the assessment and interventions. Feedback interviews will also provide information about delivery of the intervention both from the perspective of the group facilitator and the experiences of the carers themselves.

### Tolerability

Proportion of carers who withdraw or decline intervention. Record of interventions declined and why.

### Integrity of the study protocol

By examining how many participants are able to complete the study, percentage of missing data, percentage of people who completed questionnaires, percentage of people who completed each outcome measures at 6 month follow-up and calculation of the cost of running the study.

### Outcome measures

Outcome measure data will be stored in a database, and data will be analysed using the statistical package STATA (version 13). The proportion of missing items will be examined. The questionnaire data will be analysed to determine the distributions of scores. The analysis will use descriptive statistics and CIs for the parameters we are estimating. The characteristics of stroke survivors and their carers will also be described using means, SD and ranges for quantitative variables and counts and proportions for categorical variables. Data will be analysed on an 'intent to treat' basis. Any changes in the planned statistical methods will be documented in the report.

**Acknowledgements** We would like to thank the project steering group: Sheila Birchall, Charlotte Davies, Christopher Greensmith, Kate Hooban, Ben Jackson, Raj Mehta and Nikola Sprigg. We would also like to thank Joanna Fletcher-Smith, Miriam Golding-Day, Oliver Matias and Helen Taylor for their assistance with conducting the study.

**Contributors** MFW, SAT, PJW, RJF and CC drafted the manuscript. MFW, RJF and CC conceived the study. MFW is the principal investigator. LC contributed to the design of the study and delivers the intervention. MFW, SAT, EK and PB developed the intervention. All authors are members of the research team involved in the running of the study. All authors commented critically on the manuscript and read and approved the final manuscript.

**Funding** This protocol is independent research funded by the National Institute for Health Research (Research for Patient Benefit Programme, Biopsychosocial Intervention for Stroke Carers (BISC), PB-PG-0613-31064). The views expressed in this publication are those of the authors and not necessarily those of the NHS, the National Institute for Health Research or the Department of Health.

**Competing interests** None declared.

**Ethics approval** East Midlands – Nottingham 2 Research Ethics Committee (14/EMI/1264).

**Provenance and peer review** Not commissioned; externally peer reviewed.

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
