## [Reviewer comments · BMJ Open]

ARTICLE DETAILS

TITLE (PROVISIONAL)	Biopsychosocial Intervention for Stroke Carers (BISC): Protocol for a Feasibility Randomised Controlled Trial (RCT)
AUTHORS	Walker, Marion; Thomas, Shirley; Whitehead, Phillip; Condon, Laura; Fisher, Rebecca; Kontou, Eirini; Benford, Penny; Cobley, Christine

VERSION 1 – REVIEW

REVIEWER	Nan Greenwood Faculty of Health, Social Care and Education St George's University of London and Kingston University UK
REVIEW RETURNED	30-Jun-2017

GENERAL COMMENTS	Overall This is a very well written and presented protocol investigating a very important topic – supporting carers of people with stroke. However, I have some reservations and points for clarification. Perhaps the most significant of these is the heavy reliance on old literature. The references therefore need checking and updating with the inclusion of newer references where more recent evidence is available. Also, I think the paper needs a more balanced introduction to stroke caring – being an informal carer is not always all bad. Also carers are very diverse – in terms of their demographics (age, gender, amount of caring, time been caring and other support) but also not all carers will be anxious/depressed etc.. This may help explain at least in part, why carer interventions so often fail to show quantifiable impact. I found the inclusion of people with stroke confusing throughout – in the text it sounds like an intervention for carers and stroke survivors but e.g. the title does not mention them. You need to explain more clearly why stroke survivors were at the intervention. More specifically P4 – Limitations. I think there are three or maybe four important limitations missing here. Firstly, the fact that carers could be caring for someone who had had a stroke from 0-12 months this is a very long time period given that their experiences can change significantly over time. Secondly, why did you just rely on your own focus groups etc.? There is a huge body of up-to-date literature that looks into carers' needs, experiences etc.? Thirdly – this is a multi-component intervention in a group setting where you yourselves recognise the impact of the groups setting. It will be very difficult to identify what aspect of the intervention (if any) has a positive impact on the participants.
--

	Also, we know that peer support can be very important to carers – is it not possible peer support will be having a significant impact on the participants? You yourselves highlight the potential benefits of this sort of support on p7. Finally, in common with much of the available research, the participants will be/are self-selecting and may not therefore be ‘typical’ of most carers - for example in wanting support and in being able to attend the training. P5 and throughout – as mentioned above - many of these references are very old. Why have you not included e.g. some of the highly relevant, much more recent systematic reviews? P 5 – can the comment about the impact of institutionalisation be amended – this has an enormous impact on all concerned – not just the NHS! P5/6 – Also as mentioned above – I was confused why stroke survivors were included in the intervention – justification for not including them is given but they are still included. This needs clarification. P6 – the finding/conclusion from Forster et al is mentioned – that immediately after stroke it may not be an ideal time for an intervention and yet it appears this time period is included here? Typo on line 28 P7 – there are some particularly old references here – e.g. Lundh (1999) and van den Heuvel et al. (2001) Justification is needed for only relying on your own focus groups etc. in developing the intervention, rather than using the large body of available literature. P8 – it would help the reader to have some idea what ‘usual care’ entailed. The tenses here seem to be inconsistent? The term stroke survivor needs to be explained earlier. P8 – were these the only exclusion criteria – surely there had to be something around communication/cognitive impairment and being able to understand and complete the rating scales? P 11 - what is meant by a ‘well-being relapse prevention plan’? P 13 – you state that every effort to locate the participants will be made for follow up. Is this realistic for a much larger scale intervention? P 14 - line 15 – part of this is unclear - should intervention read ‘invitation’?
--	--

REVIEWER	Reg Morris Cardiff University/Cardiff and Vale UHB, Wales
REVIEW RETURNED	13-Jul-2017

GENERAL COMMENTS	This protocol makes an excellent case for conducting research into carer support following stroke. It is generally well-presented and clearly structured with appropriate attention to the key stages of the evaluation process and their attendant issues. The following points should be taken into account to improve the impact of the report and the potential utility of the proposed intervention. The proposal is to determine the feasibility of an RCT of a biopsychosocial intervention for carers within one year of stroke. The authors cite the negative finding of Forster et al.
---

(2013) for not doing the intervention sooner after stroke. However, the first year is still too soon. Personal accounts and studies of dyadic adjustment after stroke suggest that salient psychological issues emerge and crystalize later in the recovery pathway. Two to three years after stroke would be a better window for this type of intervention.

The intervention will be given to initial groups of five, but it is stated that it may be delivered to smaller numbers or even 1 to 1 if required. It is not clear why the initial group size is so small, since the intervention seems to be principally psychoeducational and capable of presentation to larger groups. (But it is not described in any detail--see below.) Groups this small will be resource-intensive and limit the affordability of the intervention within clinical services. If there is a rationale for such a resource-intensive intervention, then it should be provided.

While this an early-stage feasibility proposal and therefore necessarily short on detail, it would be helpful to have more information about the intervention in addition to information about its intended benefits (p.9). Stroke carers display a wide range of psychological difficulties, but most evidence-based psychological interventions are symptom or condition-specific. The article should indicate how the requirement to encompass a range of symptoms/conditions will be addressed. The authors state on page 9 that the intervention will be facilitated by a psychologist with 'accredited training'. This is laudable; if the intervention is to be capable of widespread adoption following evaluation it must be evidence-based, practicable, relevant to the multifaceted needs of stroke carers and have a credible theoretical underpinning.

Therefore the authors should describe the intervention they have in mind and how it achieves these requirements and matches the proposed 'accredited training'--- this will necessarily need to use one of the relatively few current accreditation providers (e.g. BABCP, BPS).

Staying on the topic of the intervention. The description of the sessions on pages 9-10 suggest that it is to be standardised. This needs clarification. Experience of stroke carer groups suggest they will bring particular issues to sessions that they want addressed. How will the protocol for sessions encompass the tension between standardisation and relevance/engagement?

I have some concerns about the proposed outcome measures in the context of a psychological intervention with a 'healthy' sample that are not selected on the basis of existing psychopathology

- Clinical anxiety and depression may not be highly elevated at baseline, engendering a floor effect. (Note: the 2013 TSA study reporting high levels of carer distress used self-reported distress, not standard measures).

- There is no measure of well-being. The absence of anxiety and depression symptoms (measured) do not imply the presence of well-being. A well-being scale should be included.

- The EuroQol is a relatively insensitive measure and does not pick up psychological change or its sequelae.

- On page 9 we are told the intervention focusses on 'adjustment to stroke', but there are no psychological adjustment measures as such. 'Hopefulness' is a good candidate as a psychological dimension sensitive to psychological intervention. Dyads in the 'usual-care' control arm will be community-living and could experience a very wide range of life-events that affect psychological adjustment.

	They could even access relevant treatments; e.g. there is now a stroke self-help book that includes carer interventions on the LTC book prescription scheme in England. The interventions proposed in this study will almost certainly overlap with those in the book. Those randomised to this arm should be asked about relevant events, including accessing psychosocial treatments. In the inclusion/exclusion criteria, please provide information about how aphasia (short of not being capable of consent) will be assessed and managed. Will SALT support be provided in the groups? On page 15, please provide more information about the principal analysis in addition to being based on 'intention to treat'. There should be an interview schedule provided for the semi-structured interview to be used in the qualitative phase of the design. The term 'nested' is used to describe the qualitative phase. This term has a particular meaning in quantitative research designs and ideally should be avoided as a description of a qualitative phase. The words 'linked' or 'associated' would be better descriptors. Summary This is a generally sound proposal that could underpin some important research. The following areas should be addressed prior to publication.  1. Reconsider the timing of the intervention. 2. Reconsider the size of the intervention groups. 3. Provide more details about the intervention in order to demonstrate its relevance, practicability, evidence-base and credibility in terms of theoretical underpinnings and alignment to accepted models. 4. The outcome measures are mental illness oriented, but the sampling is not. These measures should be supplemented (replaced?) by those more sensitive to psychological interventions and meaningful psychological change—i.e. adjustment-focussed. 5. The study should consider experiences of those in the care as usual arm that might impact on psychological adjustment.
--	--

VERSION 1 – AUTHOR RESPONSE

Reviewer One

- We accept the point about the reliance on old literature and have amended the introduction to include newer references including references to the recent systematic reviews.
- A change has been made to provide a more balanced introduction to stroke caring on page 4.
- Stroke survivors are not included in the intervention, the intervention is delivered only to the stroke carers. We seek consent from the stroke survivors because we collect data from them at baseline and follow-up. Further clarification has been provided throughout the manuscript to remove references to the recruitment of 'dyads'.
- The time-period for recruitment and delivering the intervention is a feasibility outcome and we are seeking to determine whether 0-12 months is feasible and acceptable.
- The intervention was developed using data from three areas: a) existing literature, b) focus group output and c) expert nominal group consensus.
- We accept the point that carers will be self-selecting and our sample might reflect those who want support and are willing to attend the groups. The number of eligible participants recruited is a feasibility outcome as outlined on page 13.

- The comment about the impact of institutionalisation has been amended.
- A description of usual care will be provided following the qualitative interviews with carers in the control group and this is detailed on page 11 in the section on feedback interviews.
- The typo on page 6 and inconsistency of tenses on page 8 have been amended.
- The term stroke survivor is widely used in stroke literature and we believe it does not require specific explanation.
- Exclusion criteria are listed correctly in the manuscript. We do not exclude people on the basis of their inability to complete the baseline and outcome measures; the feasibility of data collection is an outcome of the study as listed on page 15.
- The term 'well-being relapse prevention plan' has been clarified on page 10.
- We believe it is common practice to make every effort to locate participants for follow-up in randomised controlled trials. However, we will record attempts made and the feasibility of doing this and have amended the manuscript (page 14) to reflect this.
- Page 14 line 15 the term intervention is correct.

Reviewer Two

- We disagree that two to three years post-stroke would be the best time point for the intervention. Forster et al concluded that the immediate aftermath in hospital was too soon but part of the aim of the BISC intervention is to prevent the onset of psychological difficulties and we believe that two to three years would be too late for this. In part, this is based on our pre-intervention focus group findings. However, our qualitative interviews with intervention participants will further focus on the appropriateness and acceptability of the timing of intervention delivery. Early indications from these interviews tell us that the timing is appropriate and in some cases intervention even earlier may have been helpful.
- The intervention is delivered in group sizes of five for the purposes of determining feasibility and also based on the practicalities of this study. Groups of five are also typical for sessions in clinical practice. However, it is possible that this could be delivered in larger groups in a bigger trial but for a small single-centre feasibility study we believe that the group sizes are appropriate.
- As detailed highlighted above, we believe that sufficient detail regarding the intervention content has been provided in this RCT protocol.
- Standardisation of intervention. The format for the intervention is a set programme with pre-determined content and topics. Within the delivery of the programme there is scope, and indeed it is encouraged, for carers to discuss their own experiences and needs around the topic.
- The outcome measures have already been agreed and have been used for all the baseline and the first follow-up visits. It is therefore not possible to change them at this point. One of our feasibility aims is to determine the suitability of outcome measures (page 7). Additionally, part of the rationale for the qualitative interviews with intervention participants is to determine the impact of the intervention and whether the outcome measures are suitable. It is possible that a wellbeing measure may be appropriate for use in a further study.
- As detailed in the response to reviewer one, interviews with participants in the control group will provide descriptions and explore experiences of usual care. We thank the reviewer for reference to his new book and we agree that this does overlap with some of the principles and techniques of the BISC intervention. As the book was only recently published (June 2017) and our study has been ongoing since November 2015 we do not believe that it will be widely used by our control group at the present time. However, we will be mindful of this when we conduct the interviews about their experiences of usual care.
- Additional information has been added on page 8 under inclusion/exclusion criteria to explain how aphasia is assessed for the purpose of determining study participation.
- For a feasibility study the principal analysis is descriptive and we believe that we have outlined this in sufficient detail on pages 14-15.
- We do not believe that an interview schedule for the qualitative study should be provided in the protocol. This will be provided in the qualitative interview findings paper.

• We do not agree that the term 'nested' is inappropriate for the qualitative study. Indeed, this is commonly used to describe qualitative studies conducted alongside trials. We have provided references from the NIHR Research Design Service[1] and a seminal BMJ paper on qualitative research alongside RCTs[2] which both use the term nested in this context.

We would like to thank the reviewers for their detailed comments and trust that our response is satisfactory. Should you require any further information or clarification please do not hesitate to contact me and I will be happy to provide a further response.

1. NIHR Research Design Service South Central. Qualitative Research Design Secondary Qualitative Research Design 2017. <http://www.rds-sc.nihr.ac.uk/study-design/qualitative-research-design/>.
2. Lewin S, Glenton C, Oxman AD. Use of qualitative methods alongside randomised controlled trials of complex healthcare interventions: methodological study. BMJ 2009;339.

VERSION 2 – REVIEW

REVIEWER	Nan Greenwood FHSCE, St George's University of London and Kingston University, Cranmer Terrace Totting, London SW17 0RE
REVIEW RETURNED	15-Sep-2017

GENERAL COMMENTS	Thank you for responding so comprehensively to my comments and you have addressed them adequately.
--

REVIEWER	Reg Morris Cardiff University/Cardiff and Vale UHB, Wales
REVIEW RETURNED	18-Sep-2017

GENERAL COMMENTS	This report describes the feasibility study leading up to a highly worthwhile RCT that has the potential to produce a major step forward in the support for stroke carers. The revision has dealt well with the majority of the points made by the reviewers through revisions, and adequately defended others. I have only two remaining concerns: 1) It is difficult to accept that there need be no mention of statistical analysis methods in a feasibility study. If the eventual RCT is to be feasible, then surely the design outlined in the feasibility study must be capable of meaningful analysis? Having reviewed RCTs with seriously sub-optimal analysis methods I would suggest that it is logical and sensible to include a note of the intended analysis method at this stage. 2) I accept that the details of the intervention will be elaborated elsewhere. But in responding to this point the authors omitted to deal with the matter of the accreditation of the training they refer to. Accredited training is a bonus in relation to fidelity, but only if the accreditation is bona fide. The accreditation body should be cited here.
--

VERSION 2 – AUTHOR RESPONSE

Responses:

1. We disagree with the reviewer that “it is difficult to accept that there need be no mention of statistical analysis methods in a feasibility study”. We refer to the CONSORT 2010 statement: extension to randomised pilot and feasibility trials which states “A range of methods can be used to address the objectives in a pilot trial. These need not be statistical” [1] (pg.17). The manuscript highlights the detail that will be provided on both the feasibility and participant outcomes. This information will be presented narratively and using descriptive statistics; we believe this is appropriate for a feasibility trial and is consistent with the guidance in the CONSORT extension statement. However, to aid clarity, we have added a sentence to the analysis section ‘acceptability of the study design’ to explain that descriptive statistics will also be presented for feasibility outcomes in this section.

2. We accept the point about the accreditation of the training and, for the sake of clarity, we have removed the word ‘accreditation’ on page 9.

If you require any further information or clarification please do not hesitate to contact me and I will be happy to provide a further response.

VERSION 3 – REVIEW

REVIEWER	Reg Morris Cardiff University/Cardiff & Vale UHB
REVIEW RETURNED	20-Sep-2017
GENERAL COMMENTS	The minor amendments I asked for have been completed.